# Towards a Quantitative Approach for Determining DAA System Risk Ratio

**Kris Ellis \* and Iryna Borshchova**

Aerospace Research Centre, National Research Council of Canada, Ottawa, ON K1A 0R6, Canada
* Correspondence: kris.ellis@nrc-cnrc.gc.ca; Tel.: +1-613-998-5522

**Abstract:** Specific Operations Risk Assessment (SORA) is a methodology developed by the Joint Authority on Rulemaking for Unmanned Systems (JARUS) for safely conducting and evaluating Remotely Piloted Aircraft Systems (RPAS) operations in specific airspace. Many regulators, including Transport Canada (TC), the civilian aviation authority in Canada, have adopted the SORA approach to guide RPAS operators in their applications for Beyond Visual Line of Sight (BVLOS) flight. Although the qualitative approach on how to assess the performance of a Detect and Avoid (DAA) system is outlined in the SORA, a quantitative and agreed-upon approach, on how to ensure that the specific DAA system meets the required Risk Ratio criteria, has yet to be established. This paper proposes a practical approach to determining the Risk Ratio, considering sensor performance, RPA maneuvering characteristics, and airspace specifics. The developed approach relies on publicly available modelling frameworks and airspace models. Illustrative examples of applying the method to determine the Risk Ratio of specific DAA systems are presented in the paper along with a discussion on the challenges of implementing SORA into BVLOS regulations for RPAS.

**Keywords:** detect and avoid; SORA; risk assessment; BVLOS

## 1. Introduction

The accelerated growth of Remotely Piloted Aircraft Systems (RPAS) in the national airspace in recent years underscores the urgent need for mechanisms to mitigate the risk of mid-air collisions to enable safe operations conducted Beyond Visual Line of Sight (BVLOS). RPAS with long-range and high endurance are attractive for conducting missions such as pipeline or hydro-corridor inspection, forest fire monitoring, or persistent surveillance. Such operations would require a high degree of automation at the platform and operational levels [1] since they would be typically conducted in non-segregated and potentially uncontrolled airspace. A crucial element of the desired automation is Detect and Avoid (DAA) capability available onboard the RPAS, which would allow the remote pilot to be aware of conflicting aircraft, and to take the appropriate action to remain well-clear and avert collisions.

DAA systems can be decomposed into two main functions, namely, (1) 'Detect' - situational awareness, determination and annunciation of traffic that may be in conflict; and (2) 'Avoid'—de-confliction maneuver execution, and determination of 'clear of conflict'. The 'Detect' function of a DAA system depends on sensor characteristics, e.g., signal-to-noise ratio of the target vs. background, false alarm rate, and processing/thresholding methods chosen, etc. The 'Avoid' function comprises the Remain Well Clear (RWC) and Collision Avoidance (CA) sub-functions [2], and depends on the RPA maneuvering characteristics, delays due to human factors (in the case if avoidance maneuver is not automatic), airspace specifics, and size of the protection volume. The RWC function assumes tactical maneuvers are performed within a timeframe nominally sufficient to coordinate with Air Traffic Control (ATC), whereas the CA sub-function is designed for urgent maneuvers performed as a last resort to prevent mid-air collisions when all other modes of separation fail.

These defined sub-functions are found in the standard DO-365 Minimum Operational Performance Standards (MOPS) for Detect and Avoid Systems [2] which was developed by the Radio Technical Commission for Aeronautics (RTCA). The DAA capability must also give the remote pilot the ability to comply with the International Civil Aviation Organization (ICAO) Annex 2 and Rules of the Air [3], as applicable, in a given airspace.

In 2019, Transport Canada (TC), the civilian aviation authority in Canada, released Part IX of the Canadian Aviation Regulations, which provides a uniform set of regulations regarding RPAS operations conducted within Visual Line of Sight (VLOS). While the current regulations have enabled greater commercial application of RPAS technology, the limitations imposed by the VLOS requirement prevent industry from fully harnessing the potential benefits of RPAS technology. To address this, TC has begun working with numerous organizations to allow for limited-scope BVLOS operations. TC have drafted an Advisory Circular (AC) 903-001 [4] on RPAS Operational Risk Assessment, which follows the methodology described in the Joint Authority's on Rulemaking for Unmanned Systems (JARUS) Specific Operations Risk Assessment (SORA [5]), to provide Canadian RPAS operators with a structured process to assess the risks associated with RPAS operations, with a specific focus on BVLOS operations. As part of this process, the operator must determine the 'Risk Ratio', which is defined as the ability of the complete, 'end-to-end' DAA system to mitigate potential collisions with conflicting traffic.

*Problem Statement:* Although the qualitative approach on how to assess a DAA system performance was outlined in AC 903-001, and in the SORA, a quantitative and agreed-upon approach to determine the Risk Ratio has yet to be established. At the time of writing, the Annexes presenting the supporting data for the SORA air risk model have yet to be published.

*Contributions:* This paper proposes a practical, quantitative approach to determine the Risk Ratio of a given DAA system, considering sensor performance, RPA maneuvering characteristics, and airspace specifics.

The paper is organized as follows: an overview of the SORA process is presented in Section 2. The proposed approach to determine Risk Ratio is described in Section 3, beginning with the determination of the detection and avoidance volumes. The requirement for, and application of airspace models is presented, and a complete example Risk Ratio calculation is established. Next, See and Avoid as a mitigation is discussed, followed by a sensitivity study of the effects of sensor Field of Regard and detection range on Risk Ratio. Section 3 concludes with an example of how a sensor's probability of detection may be accounted for. Sections 4 and 5 present discussions and conclusions respectively.

## 2. SORA Overview

SORA [5] is a methodology developed by JARUS for assessing the risk of an RPAS operation. Essentially, SORA aims to guide both RPAS operators and regulators on how to evaluate the safety of a specific RPAS operation in a specific airspace.

The SORA definition of risk is borrowed from SAE ARP 4754A/EUROCAE ED-79A [6]; "the combination of the frequency (probability) of an occurrence and its associated level of severity". SORA focuses on the assessment of ground and air risk separately, and divides them into classes:

1.  Four Air Risk Classes (ARC): ARC-a thru ARC-d, where ARC-a is generally defined as airspace where the risk of collision between an RPAS and traditional aircraft is acceptable without the addition of any tactical mitigation. ARC-b, ARC-c, ARC-d generally define airspace with increasing risk of mid air collision. ARC can be considered as a generalized qualitative classification of the rate at which a RPA would encounter traditional aviation in the specified airspace environment.

2.  Ten Ground Risk Classes (GRC): 1 thru 10, with increasing risk of persons on the ground being injured by the RPAS.

The SORA process requires the applicant to establish the intrinsic (initial) ground and air risk classes associated with the planned operation, and then to establish the effectiveness

of the strategic and tactical mitigations (SORA Annex C [7] and D [8], respectively). This paper focuses on the mitigations for reducing the risk of mid air collisions (i.e., reducing Air Risk Class).

Strategic mitigations reduce risk by implementing modifications to the operation prior to take-off, and do not require a mitigating feedback loop (i.e., real-time de-confliction based on sensed/observed data)—for example, conducting operations during the least "busy" hours, or within certain geographical boundaries. Strategic mitigations can be used as a basis for requesting that the operation be considered at a lower ARC than its intrinsic assessment, in what is known as the Residual ARC. Ostensibly, the Residual ARC reflects the likelihood of an airborne encounter between the RPAS and traditional aviation for the specified operation.

Tactical mitigations, conversely, are applied after takeoff to mitigate residual risks, and require a feedback loop to respond to real-time observed (VLOS) or detected (BV-LOS) threats to separation provision and collision avoidance. Ostensibly, these tactical mitigations constitute the DAA system. The effectiveness of these tactical mitigations is defined by a quantity known as the Risk Ratio (RR); the ability of the complete, 'end-to-end' DAA system to mitigate potential collisions with conflicting traffic. A lower RR means more potential collisions will be mitigated, e.g., a Risk Ratio of 0.1 indicates that, out of 100 potential collisions, the DAA system would mitigate 90.

SORA introduces the concept of "robustness" levels (Low, Medium, and High) which are established as a combination of the "level of integrity" (i.e., safety gain) provided by each mitigation, and the "level of assurance" (i.e., method of proof) that the safety gain has been achieved. For a low level of assurance, the applicant may simply declare that the required level of integrity has been achieved, whereas, for a high level of assurance, the achieved integrity may have been found acceptable by a competent third party. The system's overall robustness is determined using rule-based approach; e.g., if an RPAS operator demonstrates a Medium level of integrity with a Low level of assurance, the overall robustness is considered as Low.

The performance requirements for the DAA system are defined by the Tactical Mitigation Performance Requirements (TMPR), which specify the performance, robustness, and Risk Ratio objective requirements for a DAA system given a residual ARC (i.e., after strategic mitigations have been accounted for). Table 1 presents the proposed requirements for both the JARUS SORA, and Transport Canada's SORA.

**Table 1.** DAA system requirements summary.

| Residual ARC | DAA Performance | DAA Robustness | DAA Risk Ratio |
|---|---|---|---|
| ARC-d | High | High | ≤0.1 |
| ARC-c | Medium | Medium | ≤0.33, or 0.3 * |
| ARC-b | Low | Low | ≤0.66, or 0.5 * |
| ARC-a | No Requirement | No Requirement | ≤1.0 |

* Values proposed in Transport Canada AC 903-001.

Figure 1 describes the air conflict mitigation process as suggested by JARUS, with the summarized process presented in the bottom row, and mitigations identified in the upper two rows. An alternative way to visualize the process is in the form of a fault-tree as shown in Figure 2, which has been adapted from a similar fault tree presented in [9]. The fault tree is not exhaustive in its treatment of possible mitigations; however, it illustrates how the SORA process relates to a target level of safety by reducing the likelihood of a mid air collision. A more detailed fault tree analysis may be found in [10]. For a mid-air collision to occur, there must be aircraft on a collision course, and a failure of all tactical mitigations as identified by the top most AND gate in Figure 2. The branch from the left side of this AND gate reflects the likelihood of a collision course trajectory developing, and is comparable to the Residual ARC after the strategic mitigations are applied. A fundamental difference between the fault tree analysis and the SORA approach is that there are only four

levels of granularity in this branch (i.e., ARC-a through ARC-d), as opposed to the infinite granularity of the fault tree's probabilistic approach. This is likely an intentional simplifying measure of the SORA process as the reliability values for strategic mitigations are difficult to establish. The SORA guidelines for strategic mitigations [7] state that the air risk assessment is qualitative in nature; however, it is to be supported by quantitative data where possible. It is the authors' opinion that the tactical mitigations branch is the most appropriate place to include quantitative data, as it requires such data for the calculation of the Risk Ratio. The fault tree in Figure 2 demonstrates how the Risk Ratio is equivalent to the likelihood of the tactical mitigations failing to be effective in preventing a collision. Guidelines for the tactical mitigations are provided in [8]; however, they provide no examples of how to determine the Risk Ratio from a DAA system's performance specifications. The fault tree in Figure 2 shows that the Risk Ratio is determined under the assumption that the DAA system is functioning at its nominal performance. Equipment failure, or loss of function of the DAA system, is treated separately in the SORA through the robustness requirements, and is shown in the fault tree as an input to the OR gate resulting in the failure of all tactical mitigations.

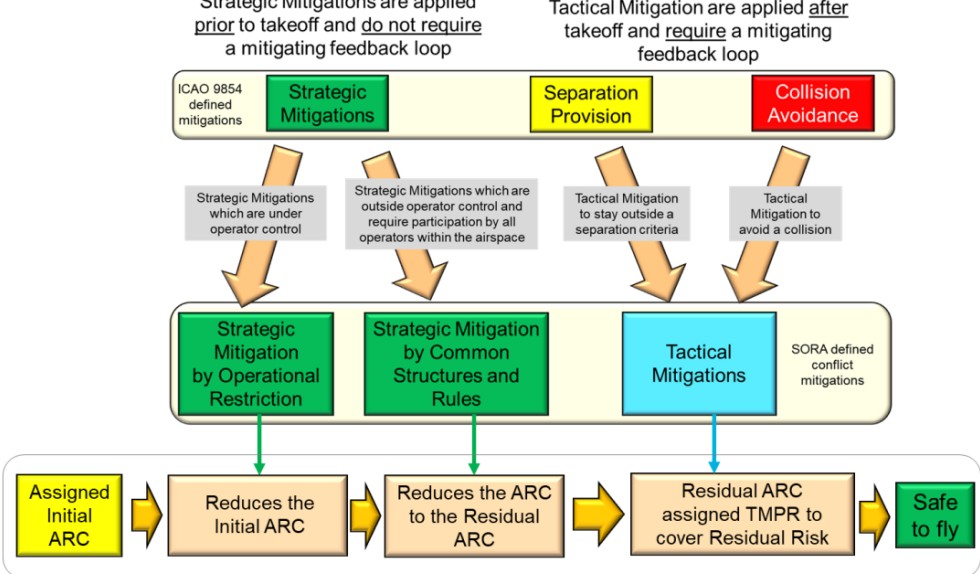

**Figure 1.** SORA Air-Conflict Mitigation Process, Credit: JARUS SORA Annex C (http://jarus-rpas.org/sites/jarus-rpas.org/files/jar_doc_06_jarus_sora_annex_c_v1.0.pdf, accessed on 3 January 2023).

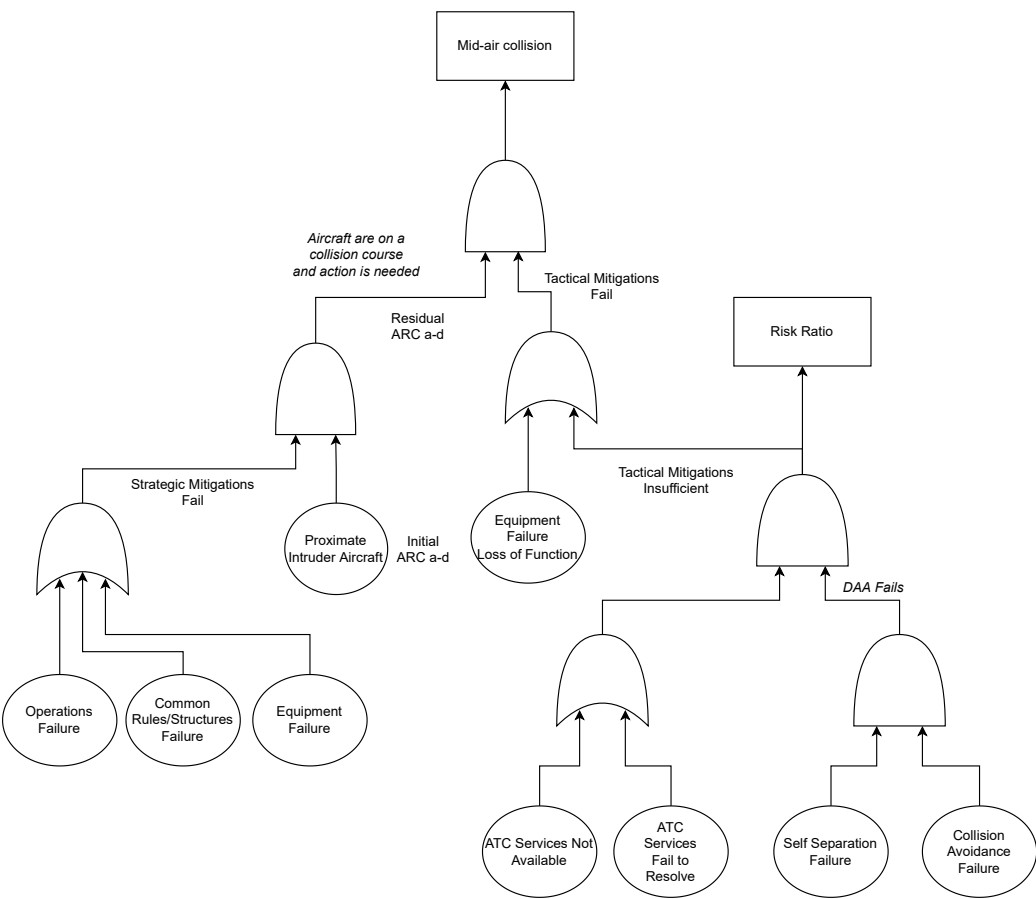

**Figure 2.** Fault Tree Presentation of ARC, Mitigations, and Risk Ratio.

## 3. Determining Risk Ratio

RPAS operators, integrators, DAA system designers, and regulators require a consistent and agreed upon method for the determination of the Risk Ratio. This section presents a proposed approach and evaluates a simplified example, along with extensions and design revisions.

### 3.1. Detection Volume

As part of its DAA performance requirements, AC 903-001 [4] defines the DAA system Detection Volume as "...the volume of airspace (temporal and/or spatial measurement) within which traditional aircraft must be detected in order to avoid a near mid-air collision, and remain well clear (if required). It can be thought of as the last point at which an aircraft must be detected, so that the DAA system can perform all the intended functions. The Detection Volume is not tied to the sensor(s) Field of View/Field of Regard (FOV/FOR). The size of the detection volume depends on the aggravated closing speed of traffic that may reasonably be encountered, the time required by the remote pilot to command the avoidance manoeuvre, the time required by the system to respond and the manoeuvrability and performance of the aircraft".

The Detection Volume is the three-dimensional space represented by the ranges, azimuths, and elevations at which the RPA must detect an aircraft in order to perform an avoidance maneuver that ensures a prescribed miss distance. Since the Detection Volume is dependent on both intruder and ownship performances (including speed, and maneuverability), there is no 'single' Detection Volume for a given RPAS and operation. Instead, there are "families" of Detection Volumes for the various combinations of ownship/intruder speeds/azimuths/elevations that may be encountered.

Ultimately, RPAS system operators/integrators, and DAA system manufacturers, are not directly concerned with the Detection Volume itself, but rather the DAA Risk Ratio for a given combination of DAA system performance characteristics, and a proposed RPAS operation. Once this Risk Ratio is established, an operator, or regulator, may then assess whether the system meets the performance specification for the Air Risk Class of that operation.

We propose a preliminary approach to establish the DAA Risk Ratio by employing a modelling and simulation framework [11] in conjunction with an uncorrelated encounter model for the airspace [12]. The range inequality relationship (Equation (1)) was developed to assist in this task [13]. This inequality represents the core relationship governing the DAA requirements: it ties into a single equation the performance requirements of the sensing system, the detection algorithms, and the intruder and ownship aircraft conflict geometries. The range inequality is generic and can be utilized to evaluate any DAA solution. Moreover, it provides a framework to which complexity can be added in a managed fashion:

$$R_0 \geq R_{det} \geq R_{warn} \geq R_{avoid} \tag{1}$$

The terms are defined as follows:

$R_0$: The range at first detection, this represents the theoretical maximum range that an intruder may be detected by the sensor system. Each sensing modality may have its own model for determining $R_0$.

$R_{det}$: The target detection range, the maximum range at which a target may be established as a collision course intruder by the processing algorithms. $R_{det}$ is a useful predictor of the processing algorithm's performance/utility. In particular, the detection range can be defined in terms of processing time, where $R_{det} = R_0 - v_{close}T_{proc}$, where $v_{close}$ is the closure rate between the ownship RPA and the intruder, while $T_{proc}$ is the processing time required to establish the detected intruder as a threat. A trade-off exists between processing time and detection range where more processing time allows for robust tracking and high target confidence, at the cost of reduced detection range, whereas shorter processing times achieve higher $R_{det}$ values at the expense of increased false-positive rates. When multiple sensors are available, each sensor will have its own $R_{det}$.

$R_{warn}$: The warning range is the minimum range at which a warning must be issued to the pilot in command (PIC) about an impending collision. The warning may also include the preferred avoidance maneuver determined by the DAA system at that point in time given available information. The warning range must take into consideration human factors interpretation and response delays.

$R_{avoid}$: The minimum range is defined as the range at which the avoidance maneuver must be initiated to ensure that the near-miss volume will not be penetrated. $R_{avoid} = v_{close}T_{man}$, where $T_{man}$ is the time required to conduct the avoidance maneuver. $R_{avoid}$ is sensor agnostic, and depends purely on the maneuvering characteristics of the RPA and the avoidance algorithm it implements.

The Detection Volume is the collection of all values of $R_{det}$ for a given RPAS operation. A comprehensive, and complete analysis would need to take into account the performance of all sensors modes, the tracking system, threat detection system, latencies, as well as human factors effects if the avoidance maneuver is to be initiated by the RPAS pilot.

In the interest of demonstrating the framework and approach for determining the DAA Risk Ratio, some simplifying assumptions are made; however, it should be understood that the complexities associated with detection/tracking performance, and uncertainties can be added to the framework later. This paper concentrates on the Avoidance Volume, rather than the Detection Volume. The Avoidance Volume can initially be thought of as the equivalent Detection Volume if the system had "perfect" probability of detection, "perfect" tracking accuracy, and zero latency.

The Avoidance Volume is useful for both system developers/integrators and regulators as it establishes an absolute minimum level of performance that must be achieved by the system. The addition of detection probability, tracking performance, etc., will only result

in larger volumes to establish the overall Detection Volume. With some assumptions (presented in Section 3.2), the Avoidance Volume can be determined using:

1. The RPA speed and collision avoidance maneuver performance;
2. Human factors, and command and control link latency in initiating the avoidance maneuver; and
3. A probability distribution of aircraft speeds in the airspace in which the operation is planned.

*3.2. Determining the Avoidance Volume*

As this approach is still preliminary, several assumptions and simplifications have been made to ensure the overall process and underlying philosophy are easily understood. The fundamental assumptions at this stage include:

1. A single non-maneuvering, level flight intruder aircraft;
2. A level flight ownship RPA prior to the avoidance maneuver;
3. A fixed volume of space around the RPA, known as the Collision Volume (CV), is protected;
4. A uniform distribution of intruder aircraft initial positions and directions;
5. A horizontal avoidance maneuver is automatically performed by the RPA, as per the algorithm described in [11].

While the present analysis is limited to the horizontal case for the sake of simplicity, it is relatively straightforward to extend the analysis to include vertical maneuvers, and climbing/descending intruders. The inclusion of manually piloted avoidance maneuvers can be considered through the incorporation of an appropriate human factors delay.

It is critical to have an agreed upon definition of the CV, as it impacts the analysis that follows. One popular definition of the CV protects a cylinder of radius 500 ft, and a height of ±100 ft centered on the RPA as shown in Figure 3.

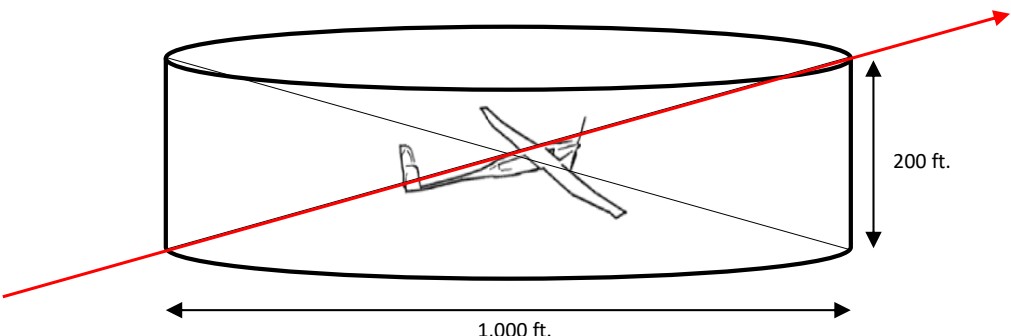

**Figure 3.** Collision volume.

While AC 903-001 [4] does not specify a particular Collision Volume, and instead refers to collisions directly, it does mention the near mid air collision boundary definition which is equivalent to that shown in Figure 3. The calculation of actual collisions would require complex three-dimensional modelling analysis, and thus an approach that uses a simple Collision Volume is preferable. Two potential approaches include:

1. Employing the CV of Figure 3, and applying a probability of collision once it is pierced [9];
2. Establishing a collision volume based on a combination of the maximum dimensions of the RPA and the nominal/mean intruder aircraft in the airspace of the operation.

For the purposes of this paper, a penetration of the 500 ft. horizontal radius and ±100 ft. vertical Collision Volume have been identified as being equivalent to a collision.

Previous research [11] outlined an approach to establish the range at which an avoidance maneuver must be performed given an ownship RPA speed, intruder speed, and a well defined horizontal collision avoidance maneuver. To determine the minimum required sensor range, the DAA system must compute a collision avoidance maneuver time, $T_{man}$, which is the minimum time by which a maximum-effort maneuver would be able to protect the CV at the time of Closest Point of Approach (CPA), e.g., 500 ft. horizontal, or $\pm 100$ ft. vertical, for a given set of RPA performance characteristics, and the relative velocity between RPA and intruder. This relationship is given by:

$$R_{avoid} = v_{close} T_{man} \qquad (2)$$

where $R_{avoid}$ is the minimum range at which a CA maneuver can be performed and still guarantee protecting the CV from penetration at the time of CPA, and can be derived from $T_{man}$ if the closing velocity between the RPAS and intruder $v_{close}$ is known. The family of $R_{avoid}$ ranges for any given azimuth/elevation, and intruder/ownship speed combination comprise the Avoidance Volume for the DAA system. A DAA sensor must demonstrate detection range performance greater than the maximum $R_{avoid}$ to successfully track the intruder over several observations and establish a high-confidence threat status. The surplus range required to establish the high confidence track forms the Detection Volume.

As an illustrative example of how to determine the Risk Ratio, a sample airborne DAA system is introduced with the following characteristics:

1. Sensor FOV—60 degrees
2. Detection range—1 km;
3. Ownship RPA speed—60 knots;
4. Automatically executed NRC Horizontal collision avoidance maneuver (constant-rate horizontal turn to a new track angle) [11];
5. Maximum bank angle—45 degrees;
6. Maximum roll rate—10 degrees per second;
7. RPA wingspan—1.5 m.

The example RPA and DAA system above is used consistently throughout this section to demonstrate the approach for determining Risk Ratio. In Section 3.5, a DAA system parametric study is conducted to establish which DAA system characteristics (e.g., range and FOV) have the greatest impact on the Risk Ratio for the given RPA speed and avoidance performance.

The horizontal avoidance algorithm evaluated the $T_{man}$ for a suite of 34 available turn options between $\pm 90$ degrees of heading change (increments of 5 degrees), conducted as a level coordinated turn with a maximum bank angle of 45 degrees, and roll rate of 10 degrees per second. The maneuver with the smallest $T_{man}$ (i.e., the maneuver that could be performed last) was selected for the determination of $R_{avoid}$. Figure 4a presents a sample of two $R_{avoid}$ vs. azimuth Avoidance Volumes for a fixed ownship speed of 60 knots and two intruder speeds (40 and 120 knots). The approach used to establish these Avoidance Volumes is detailed in [11]. Azimuth, in this context, refers to the angle between the velocity vector of the RPA and the intruder. In the figure, the blue solid line shows the 120 knots intruder case, in which all collision trajectories are considered as 'on coming'. For the 40 knots intruder condition, it becomes possible for the RPA to overtake the intruder. Here, the 'on coming' cases are shown by the black dashed line, whereas the cases where the intruder is being 'overtaken' are shown by the orange dotted line. This plot shows that, for the 40 knots intruder condition, a sensor FOV covering $\pm 40$ degrees would cover all azimuths, whereas $\pm 180$ degrees would be required to mitigate all possible collision azimuths with the faster 120 knots intruder.

Figure 4b shows closing velocity as a function of azimuth for the same conditions. This plot shows a consistent result with the $R_{avoid}$ plot of Figure 4a, with the maximum angle subtended by the origin, and the 40 knots closing velocity circle being approximately $\pm 40$ degrees.

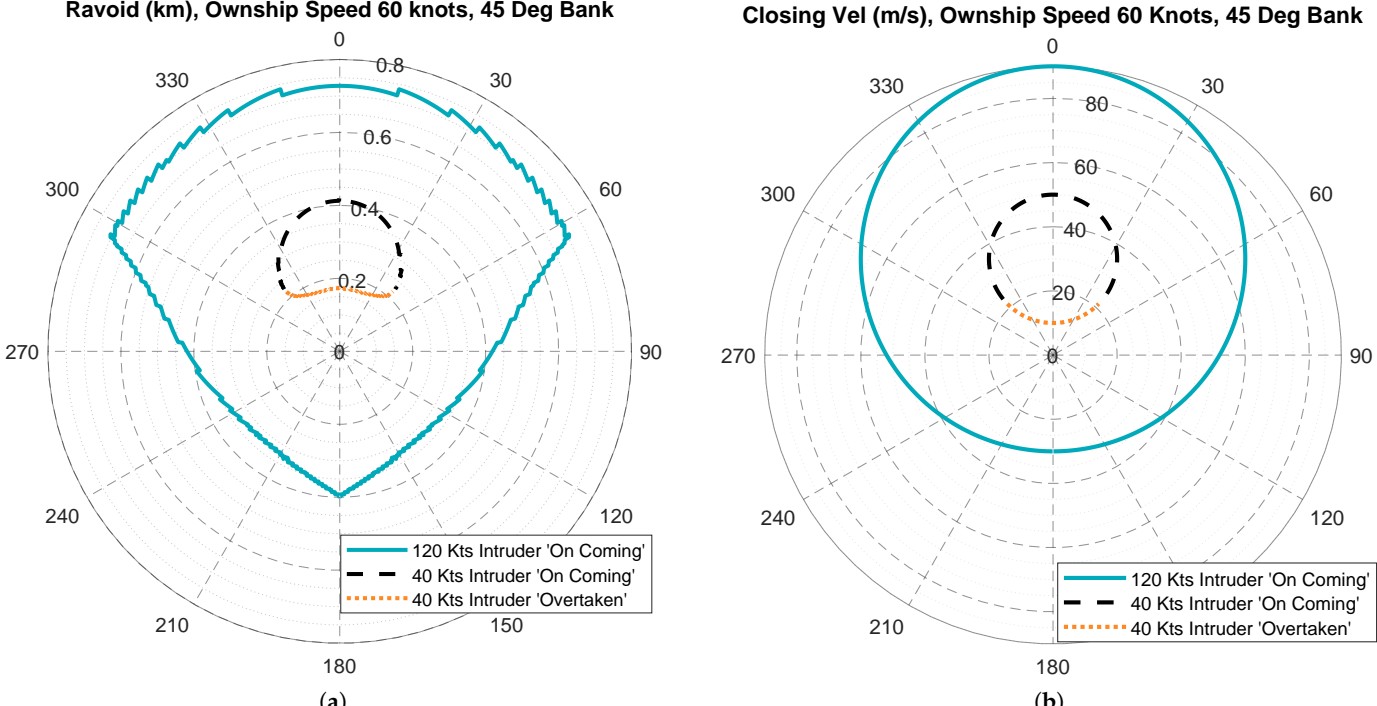

**Figure 4.** $R_{avoid}$ and Closing velocity for a fixed ownship speed of 30 m/s and intruder speeds—20 and 60 m/s. (**a**) $R_{avoid}$ vs. Azimuth; (**b**) closing velocity vs. Azimuth.

Figure 5 presents a complete analysis of the ratio of CV penetrations to mitigated collisions for the 40 and 120 knots intruders relative to the 60 knots ownship RPA for the example DAA system with a 60 degree Field of View (FOV) and a 1 km range sensor. The dashed lines illustrate the FOR (i.e., FOV and range) of the DAA sensor. If $R_{avoid}$ for a particular azimuth is within the sensor FOR, then the DAA system would mitigate this collision case, assuming a 100% probability of detection and instantaneous track establishment. Each azimuth was evaluated at a one-degree resolution, with cases where $R_{avoid}$ falls within the sensor FOR and range identified as 'passes', shown by the solid green line in Figure 5, and all other cases identified as 'fails' shown by the dashed red line. The Risk Ratio for this particular combination of ownship and intruder speed $RR_{v_i}$, evaluated in one-degree increments, can then be calculated as:

$$RR_{v_i} = \sum_{Az=0\ Degrees}^{359\ Degrees} Fails/360 \qquad (3)$$

Using Equation (3), Figure 5 shows that, for a 40 knot intruder, the Risk Ratio is 0.15 (i.e., 15% of cases fail; only 15% of cases have an $R_{avoid}$ that is outside the sensor field of regard), whereas, for a 120 knot intruder, the Risk Ratio is 0.83 (i.e., 83% of cases fail).

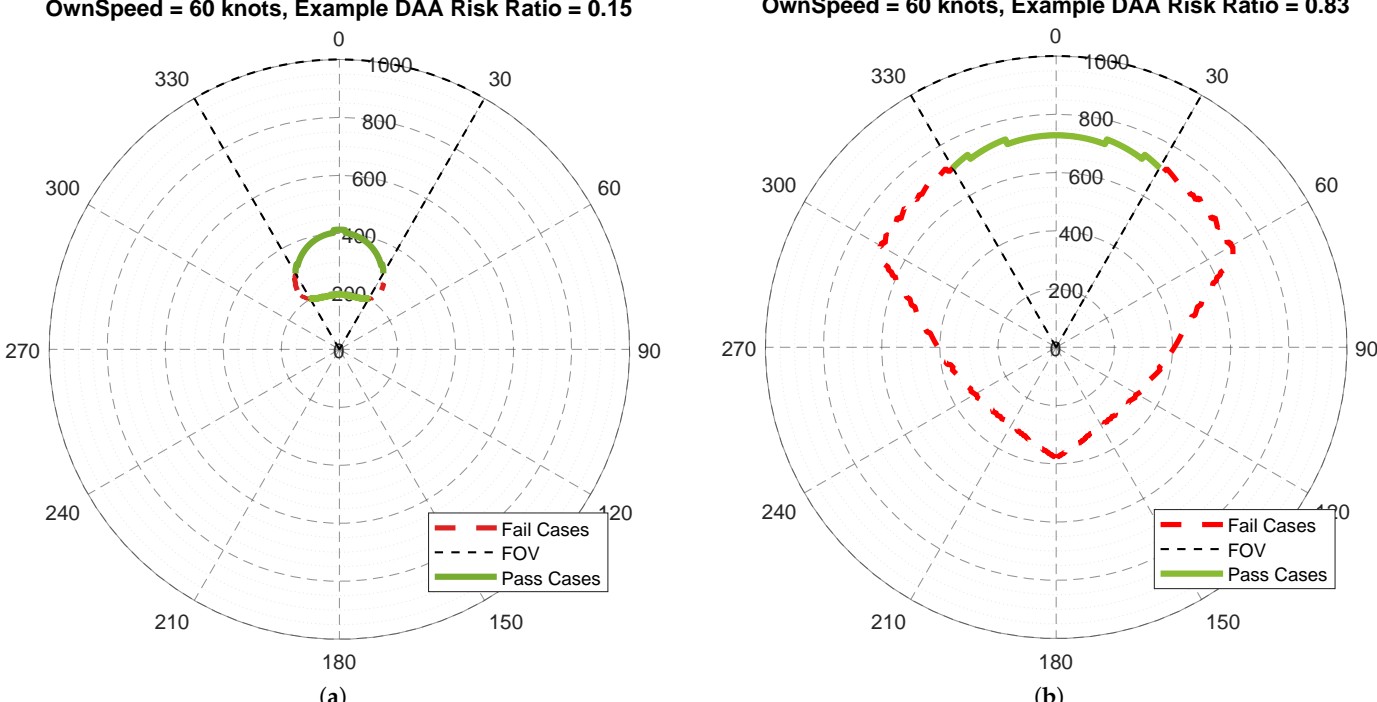

**Figure 5.** Determining the ratio of CV penetrations to mitigated collisions for a 60 knots ownship vs. 40 and 120 knots intruders. (**a**) 40 knots Intruder, RR = 0.15; (**b**) 120 knots Intruder, RR = 0.83.

### 3.3. Airspeed Probability Distribution Model

Section 3.2 presented two examples of Risk Ratio calculation for two intruder speeds—40 and 120 knots. To calculate the complete Risk Ratio of the DAA system, all possible combinations of intruder speeds, and their relative likelihood is required; i.e., statistical airspace models are required.

Figure 6 presents a sample statistical model which is assumed to represent the typical intruder airspeed distributions for the airspace where the RPAS BVLOS mission is to be conducted (e.g., below 10,000 ft). Such models have been developed for the United States NAS by MIT Lincoln Labs [12]. The validity of the Risk Ratio determined by the method proposed in this paper is highly dependent on the accuracy/applicability of the airspace model. To this end, the National Research Council of Canada (NRC) is presently working in collaboration with TC, Carleton University, and NAV Canada on the development of statistical airspace models specific to Canadian airspace, and Canadian RPAS operations.

The airspeed probability curve can be discretized into bins with a fixed width (for example 10 knot increments) to reduce the computational burden, and to improve the interpretation of results. The probability density function of Figure 6 is normalized such that its cumulative sum is 1; therefore, the magnitude of each velocity bin reflects the probability of encountering traffic at that speed. In this simplified analysis, it is assumed that the distribution of collision geometry azimuth is random and uniform.

As was shown in Section 3.2, one can use Equation (3) to determine the Risk Ratio for a single intruder and ownship velocity pairing, $RR_{v_i}$. While this ratio is of value for assessing the performance of a single velocity bin, the normalized values of 'passes' and 'fails' *over the full range of possible intruder velocities* must be used to establish the total system performance.

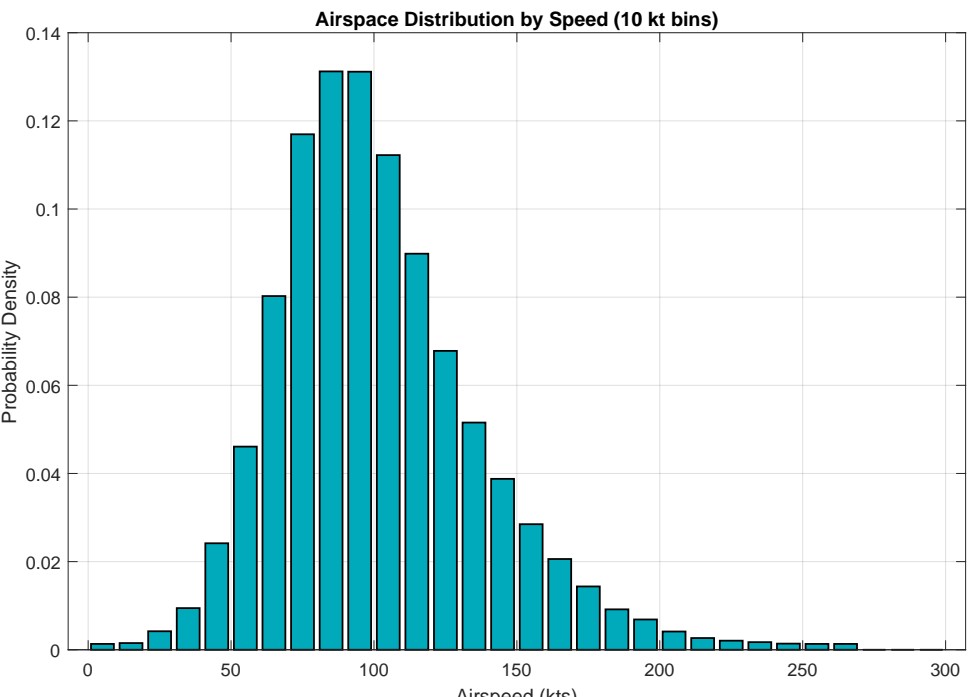

**Figure 6.** Sample mixed airspace model—airspeed distribution (30 bins from 0 to 300 knots).

The example in Figure 5 only showed two intruder speed bins (40 and 120 knots). To account for the complete distribution of aircraft speeds, one must take the Risk Ratio for the speed bin, $RR_{v_i}$, and multiply it by the height of the bin in the normalized probability distribution shown in Figure 6. This combination establishes the DAA Risk Ratio contribution for this bin. The process outlined above must be repeated for each bin in the probability distribution until the complete DAA risk ratio is determined from the cumulative sum of the individual bins as:

$$DAA\ Risk\ Ratio = \sum_{v_i bin=1}^{N_{Velocity\ Bins}} RR_{v_i} P(v_i) \tag{4}$$

Here, $i$ represents the velocity bin number from Figure 6, and $P(v_i)$ is the normalized probability of encountering traffic with intruder velocity equal to $v_i$ (the bin velocity).

Figure 7 shows the Risk Ratio analysis of our example DAA system described in Section 3.2, with a 1 km range and 60 degrees FOR, and the airspace distribution of Figure 6. The upper plot shows the individual pass/fail ratio multiplied by the probability distribution $RR_{v_i} P(v_i)$ for each velocity bin, with the lower solid red bars representing the cases that fail to be mitigated by the DAA system, and the stacked solid green bars representing the cases that were successfully mitigated. The ratio of the height of the red bar to the total height is the Risk Ratio for the individual bin, i.e., $RR_{v_i}$. In this plot, it can be seen that $RR_{v_i}$ increases dramatically once the intruder speeds are faster than the RPA ownship speed of 60 knots.

The lower plot of Figure 7 shows the cumulative contribution of the intruder velocity bins, with the red bars representing the proportion of cases that fail to be mitigated by the DAA system. Here, it can be seen that, once the final velocity bin has been accounted for, the total bar height is 1, and the Risk Ratio of the example DAA system is determined to be 0.82; equivalent to the final height of the right-most red bar. For the given airspace distribution, the example RPA plus DAA system combination is insufficient for operations above ARC-a.

The authors note that this analysis is based on CV penetrations, which are likely to be more frequent than collisions (i.e., not every penetration of the CV should result in a collision), thereby driving the DAA Risk Ratio artificially higher. The authors expect that this issue will be clarified as regulatory language surrounding the application of SORA evolves.

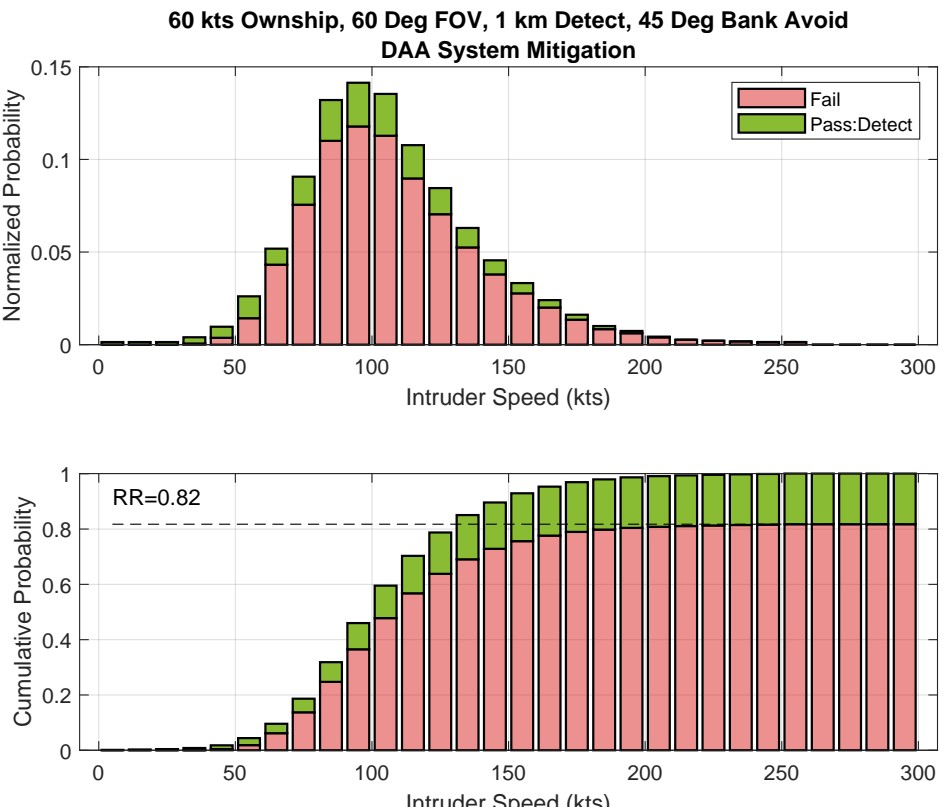

**Figure 7.** RR for the example DAA system.

### 3.4. Accounting for See and Avoid

While the DAA Risk ratio aims to assess the effectiveness of the DAA system itself, the ability of the human pilot to see and avoid collisions should not be ignored. Taken to an extreme as a "thought experiment", a low-speed, large airship RPAS is unlikely to encounter mid-air collisions owing to its ease of being seen, and low closure rate.

To establish the ability of the intruder to detect an RPAS, the following factors, at a minimum, should be considered:

1. The size of the RPA;
2. Whether the RPA is detected above or below the horizon (penalty factor for below the horizon due to ground clutter);
3. Closure rate between the RPA and intruder;
4. The Field of View from the intruder aircraft.

Modeling the ability of the intruder pilot to detect an RPAS can be treated in the same manner as modeling any other sensor for airborne target detection. MIT Lincoln Labs has developed a mathematical model of air-to-air visual acquisition under daylight conditions [14]. The original model was notably used to analyze the 1986 Aeroméxico Flight 498 midair collision over Cerritos between, a large fixed-wing multi-engine and a small fixed-wing single engine aircraft. MIT Lincoln Labs have implemented this surveillance model as a module for their DAA Evaluation of Guidance, Alerting, and Surveillance (DEGAS) framework which is publicly available.

In this paper, the authors present a simplified approach to model the ability of the intruder pilot to see and avoid an RPAS as an illustrative, and easily understood straw man. For practical applications, the use of comprehensive models, such as the MIT Lincoln Labs DEGAS, are recommended.

The simplified approach is to determine an angular resolution minima where the pilot would be able to see the RPA with high confidence, as well as a time threshold for executing an avoidance maneuver. For example, it could be assumed that, if the pilot sees the RPA for 12.5 s [15], this should be sufficient time for the potential collision to have been averted by the pilot. Ref. [16] suggests that:

1. Targets with visual angles less than one arc minute are unlikely to be seen;
2. Targets with visual angles greater than 10 arc-minutes are likely to be detected (but not necessarily recognized);
3. Targets become recognizable between 30–40% of the time when they render a visual angle of 15 arc minutes or more;
4. In four of the six models, targets become recognizable 50–100% of the time when the visual angle exceeds 30 arc-minutes.

From this research, the range at which one can declare the RPA detected is:

$$R_{det} = \frac{d}{2} \cot\left(\frac{\theta}{120}\right) \tag{5}$$

where $\theta$ is the angular resolution of the threshold of detection, in arc-minutes, and $d$ is the critical dimension of the RPA. The research from [16] applies to ground targets within clutter, whereas Ref. [14] suggests that a minimum visual angle of 2 arc minutes must occur prior to an airborne target being observed. A threshold $\theta$ of 10 arc minutes was selected as a conservative estimate, using Equation (5) with a critical RPAS dimension of 1.5 m, and $\theta$ of 10 results in an $R_{det}$ of 515.7 m. One can then use an avoidance time model for the human pilot (e.g., 12.5 s [15]) and determine a maximum closing rate under which the CV should not be pierced owing to See and Avoid. For our CV of 500 ft (152.4 m), $R_{det}$ of 515.7 m, and 12.5 s avoidance model, the following closure rate is obtained:

$$V_{See\ and\ Avoid} = \frac{(R_{det} - CV)}{T_{avoid}} = \frac{(515.7\ \text{m} - 152.4\ \text{m})}{12.5\ \text{s}} = 29.1\ \text{m/s} \tag{6}$$

Equation (6) implies that the intruder pilot could mitigate any collisions with a closing speed of less than 29.1 m/s (56 knots). Figure 8 presents an example of including See and Avoid within the RR calculation for the example DAA system with an FOR of 60 degrees and 1 km range. To prevent double accounting in this analysis, it is imperative that the cases mitigated by See and Avoid only come from the cases that otherwise failed when the DAA system was considered by itself (i.e., the red bars from Figure 7). In this example, it can be shown that See and Avoid is only a successful mitigation below intruder speeds of 110 knots, and the combined effect of DAA and See and Avoid only result in a Risk Ratio of 0.69, which is still insufficient for operations in airspace above ARC-a. It should be noted that Equations (5) and (6) did not consider the FOV from the intruder pilot perspective. A comprehensive analysis would need to consider the intruder aircraft type as well as the applicable standards for cockpit FOV. As such, the figures including the effects of See and Avoid in this paper should be considered as permissive.

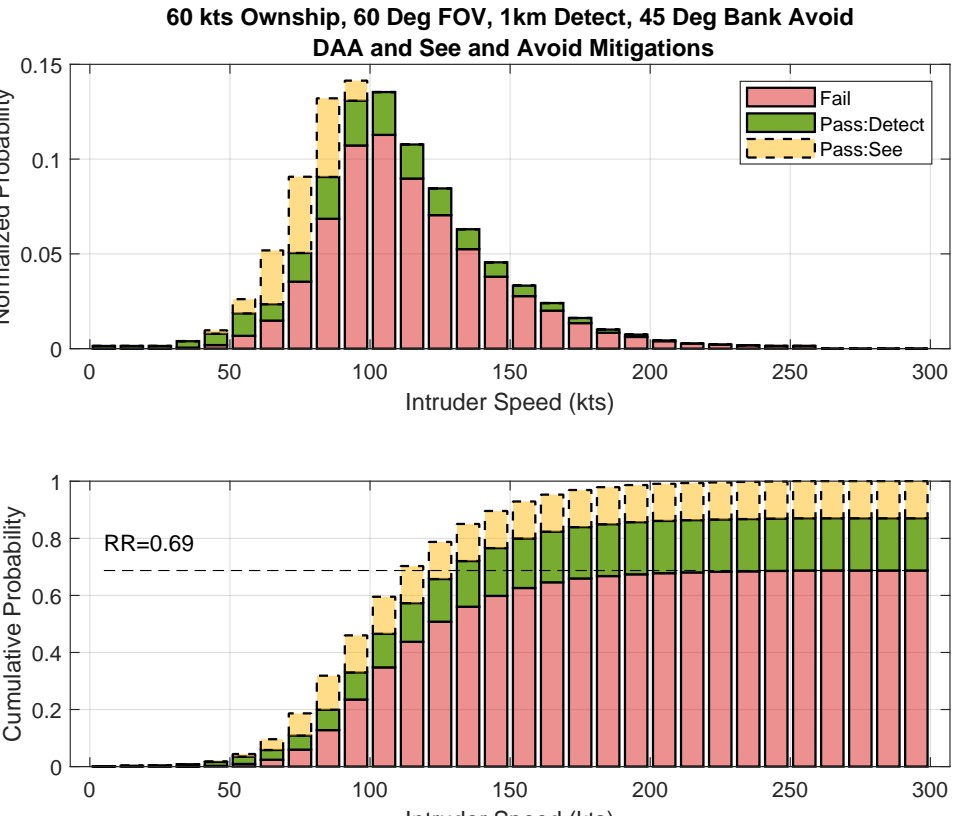

**Figure 8.** RR for the Section 3.2 example DAA system including "See and Avoid" as a mitigation.

The authors recognize that See and Avoid may be considered as a strategic, as opposed to tactical, mitigation by regulatory agencies; however, the approach of determining its efficacy via the Risk Ratio calculation may still prove of value as a means of demonstrating its effect in a quantitative manner. For example, if the applicant was attempting to justify a reduction in ARC from an initial ARC-c to a Residual ARC-b through exploitation of See and Avoid alone, the effectiveness of See and Avoid should be comparable to the difference in DAA Risk Ratio between the initial ARC and the Residual ARC (0.33 for the JARUS SORA [5], or 0.2 for Transport Canada [4]), as per Table 1. This approach may also allow for the consideration of partial credit for See and Avoid as a means of reducing the Residual ARC.

### 3.5. DAA System Design Parameter Sensitivity

Despite factoring See and Avoid, the initial example DAA system described in Section 3.2 was insufficient for operations in airspace above ARC-a. This section examines the impact of expanding the FOV and detection range on the DAA Risk Ratio.

Increasing the FOV to 120 degrees, and detection range to 2 km, values typical of current small radar sensors result in a net change in Risk Ratio as shown in Figure 9. The resultant Risk Ratio of 0.52 is sufficient for ARC-b operations based on the JARUS SORA [5]; however, it is slightly above the ≤0.5 value proposed by Transport Canada.

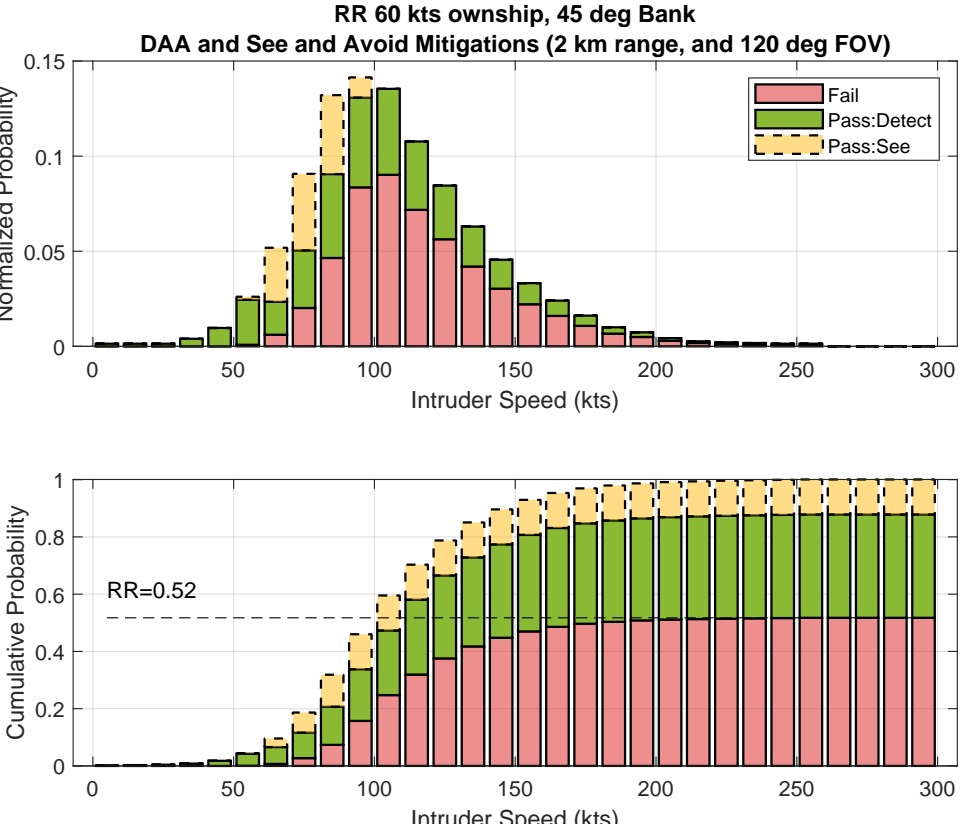

**Figure 9.** Risk Ratio of the Section 3.2 example DAA system after increasing FOV to 120 degrees and range to 2 km.

A parametric study was conducted to further explore the sensitivity of Risk Ratio to variations in FOV and detection range for the RPA characteristics detailed in Section 3.2. This type of analysis allows for the system integrator to establish what combinations of sensor performance will meet the Risk Ratio requirements for the ARC of the desired operation. FOV was varied in 5 degree increments from 5 to 360 degrees, and detection range was varied in 50 m increments from 50 to 3000 m, resulting in the Risk Ratio plots shown Figure 10, where the contour lines represent Risk Ratio increments of 0.1. The original system FOR of 60 degrees/1 km is shown by the blue X in Figure 10b, and the 120 degrees/2 km system is shown by the red triangle. The Risk Ratios shown in the figure include the contributions from See and Avoid as per Section 3.4. An interesting observation can be made: increasing sensor range has minimal impact on the DAA system Risk Ratio beyond a certain value. The results show that there is a definite 'floor' to the Risk Ratio that appears to be established by the FOV: beyond a certain range (e.g., 1000 m), there is no benefit to Risk Ratio in increasing detection range. The results also show that, for the example RPA to operate in ARC-d with a RR of ≤0.1, an FOV of at least 300 degrees is required.

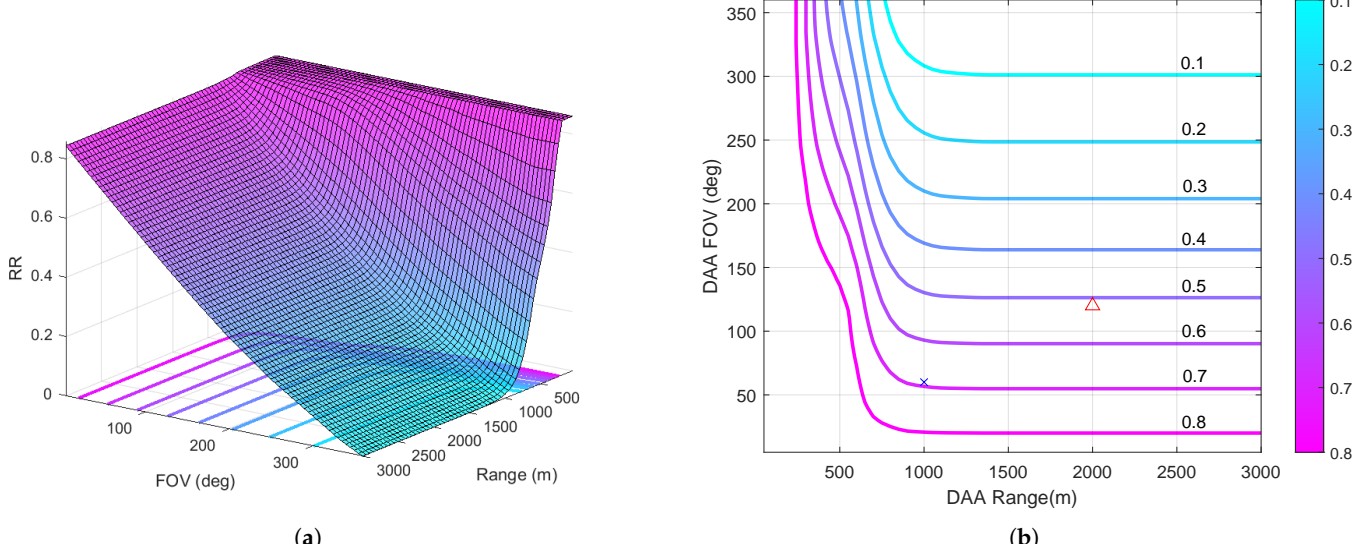

**(a)**　　　　　　　　　　　　　　　　　　**(b)**

**Figure 10.** Risk Ratio Sensitivity to Sensor Range and FOV for a 60 knots RPA, and 45 deg bank, including See and Avoid. (**a**) carpet plot; (**b**) contour plot.

### 3.6. Accounting for Probability of Detection

The RR determination approach thus far has only considered detection of the intruder aircraft as a discrete boolean condition; i.e., if the intruder aircraft was within the FOR of the DAA system, it was deemed to be detected, whereas, if it was outside, it was deemed to be missed. This is a reasonable starting point, and a valid approach for considering the sensor FOV; however, in practice, there is typically a range wise variation in the probability of a target being detected. This probability needs to be factored into the overall calculation of Risk Ratio.

Figure 11 presents Signal to Noise Ratio (SNR) and probability of detection curves as a function of range for two target types (General Aviation target with a RCS of 3 m$^2$, and an airliner target with an RCS of 50 m$^2$), and a sample radar with the following characteristics:

$P_t$: transmit power, 2 Watts;
$G$: antenna and receiver gain, 60 dB;
Frequency, 24 GHz;
$\tau_p$: Pulse width, 0.4 µs;
$T_s$: Effective antenna temperature, 300 K;
Probability of false alarm, $10^{-6}$;
Number of pulses, 25.

The signal to noise ratio, assuming no losses, can be estimated [17] using:

$$SNR = \frac{P_t G \lambda^2 \sigma \tau_p}{(4\pi)^3 R^4 k T_s} \tag{7}$$

where $R$ is range, $\lambda$ is the wavelength of the radar, $k$ is Boltzmann's constant, and $\sigma$ is the RCS of the target. The probability of detection curves of Figure 11 were determined using the method described in [18] using a Swerling 1 target type. Once the probability of detection as a function of range $P_d(R)$ is known, it can be evaluated at $R_{avoid}$ for each intruder and ownship speed combination. Factoring this into Equation (3) results in:

$$RR_{v_i \sigma} = \sum_{Az=0 \; Degrees}^{359 \; Degrees} P_d(R_{avoid}, \sigma)/360 \tag{8}$$

where $RR_{v_i\sigma}$ is the risk ratio for a given intruder velocity, and RCS ($\sigma$), $P_d(R_{avoid}, \sigma)$ is the probability of detection at $R_{avoid}$ given an RCS of $\sigma$. If the azimuth is outside of the sensor FOV, then $P_d(R_{avoid}, \sigma)$ is set to zero. Figure 11 also introduced two target sizes, namely General Aviation, and Airliner. Each target size has its own probability of detection curve. If the traffic distribution of the aircraft types is known as part of the airspace model, it then becomes possible to account for this difference in the risk ratio as follows:

$$RR = \left(\sum_{\sigma 1, v_i bin=1}^{N_{Velocity\ Bins}} RR_{v_i\sigma_1} P(v_i) P(\sigma_1)\right) + \left(\sum_{\sigma 2, v_i bin=1}^{N_{Velocity\ Bins}} RR_{v_i\sigma_2} P(v_i) P(\sigma_2)\right) \quad (9)$$

where $RR_{v_i\sigma_1}$ is the Risk Ratio for a given intruder velocity and RCS ($\sigma 1$), and $P(\sigma_1)$ is the normalized probability of encountering that intruder RCS. Equation (9) above has been written out for two intruder RCS characteristics (e.g., General Aviation and Airliner); however, it can be readily expanded to the number of traffic types contained in the airspace model. Unfortunately, the simple mixed airspace model shown in Figure 6 does not breakdown the traffic speeds by aircraft type, precluding the ability to provide an example application of this approach.

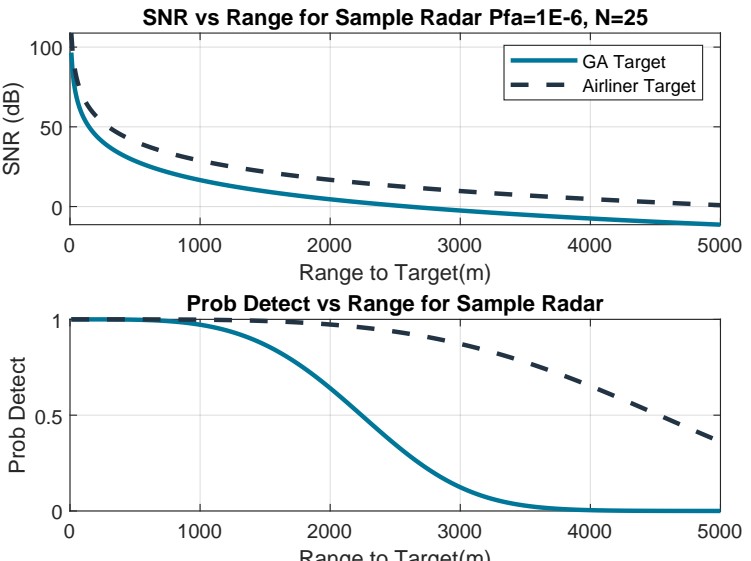

**Figure 11.** SNR and probability of detection for sample radar and two target types.

## 4. Discussion

This paper proposed a practical approach to determining the Risk Ratio based on DAA system, RPA performance, and airspace specifics. Several simplifying assumptions were made in the explanation of the approach and development of the Risk Ratio for the example DAA system. The present analysis was limited to horizontal maneuvers, and level flight intruders; however, a complete analysis must consider vertical maneuvers as well as climbing/descending intruders. The inclusion of vertical avoidance maneuvers is a relatively trivial addition to the modelling framework, whereas the consideration of climbing/descending intruders requires the addition of climb rate bins to the Risk Ratio calculation of Equation (4), and correlated climb/descent statistics to the airspace distribution of Figure 6.

The approach presented only considered a single mode of detection, namely a non-cooperative system with a defined FOR and sensor range. It is likely that many DAA systems will consist of multiple detection modes; for example, Automatic Dependant Surveillance—Broadcast (ADS-B) in combination with a non-cooperative sensor. For these cases, the RR contributions from each system may be combined in aggregate, provided that

"pass" conditions are only counted once per bin (i.e., a "pass" is an OR wise combination of all available tactical mitigations), in a manner similar to how See and Avoid was included as shown in Section 3.4. For mitigations requiring intruder aircraft equipage (e.g., ADS-B), there must be airspace models supporting the ability to determine the probability of equipage as a function of speed and climb rate, or aircraft type.

Although the paper was only able to present limited examples of Risk Ratio determination owing to space constraints, the following key factors have been empirically observed:

1.  Avoidance Algorithm and RPA maneuvering performance—these characteristics establish the families of $R_{avoid}$ curves against which the sensor range and FOV may be evaluated, and as such play a dominant role in the system Risk Ratio;
2.  RPA speed—slower RPAs will require greater sensor FOV relative to faster RPAs, while faster RPAs will require greater sensor range. At slower speeds, vertical maneuvering may be preferred for VTOL RPAs, owing to the collision volume being smaller in the vertical direction. The speed of the RPA may also change over the course of a mission. The effect on Risk Ratio may be accounted for by partitioning the mission duration into different speed regimes, evaluating each regime separately, and combining into an aggregate by weighting based on time spent in each regime;
3.  RPA size—larger RPAs increase the range of detection for "See and Avoid". The larger size will likely also result in a higher Ground Risk Class;
4.  Sensor detection range in excess of $R_{avoid}$—can be used to improve the probability of detection. A sensor with a theoretical $R_0$ detection range well in excess of $R_{avoid}$ is likely to have improved probability of detection at $R_{avoid}$. It is important to note that, although this analysis did not consider the effect of false positives, manufacturers and system integrators must strive to strike the appropriate balance between detection range sensitivity and the likelihood of false alarms.

The development of the approach to calculate Risk Ratio raised potential areas of concern for regulators, standards bodies, manufacturers, and academia:

1.  The notion of a Collision Volume is not defined in TC's AC 903-001. It is a notable absence, and could lead to each DAA system developer/integrator adopting their own definition for what volume is protected by their system. The authors note that, in the JARUS SORA Appendix I [19], the Risk Ratio definition directly refers to Near Mid Air Collisions (NMAC) with a Collision Volume definition consistent with that of Figure 3. It is recommended that TC adopts an NMAC Risk Ratio definition consistent with the JARAS SORA;
2.  While a target DAA Risk Reduction ratio of 0.5 may seem easy to meet at the outset, the authors suspect there will be significant challenges as there appears to be an implicit relationship between the Risk Ratio requirement and the Field of View of the sensor for slower moving RPAs. Taken to an extreme, one could see that even a "perfect" DAA system (infinite range) would require an FOV of at least 324 degrees to achieve a Risk Ratio of ≤0.1 if the RPA hovers at 0 m/s velocity. In the hover condition, all cases are 'on-coming' with constant closure rate equal to the intruder's velocity; thus, to mitigate 90% of all cases, a 90% FOV (324 degrees) is required. It is believed that this relationship between Risk Ratio and FOV will vary as a function of RPA speed, with the higher FOV requirements being for the slower RPAs. Conversely, faster RPAs may require less FOV; however, a longer range to ensure that avoidance maneuvers can be conducted within sufficient time;
3.  The effectiveness of the developed method heavily relies on the quality of available models. Standards bodies such as ASTM F38 WG 62669 on Testing Methods are currently exploring modeling and simulation as a primary approach to verify whether a DAA system meets the RR criteria, since flight testing a DAA system is expensive and requires experience and expertise to safely conduct collision intercepts [20,21]. This necessitates generically applicable, community-supplied, and well-understood models being publicly available, such as those identified in Annex A;

4. Is "See and avoid" a tactical or strategic mitigation? The approach presented in this paper includes "See and avoid" in RR reduction (tactical mitigation). However, one can argue that making an RPAS of brighter color or equipping it with lights to make it more visible to human pilots takes place prior to take-off and could be interpreted as a strategic mitigation for ARC reduction. However, there are only four ARC levels defined in the SORA, requiring the "See and avoid" safety gain to be significant if it is to reduce the residual ARC level. Including "See and Avoid" into the Risk Ratio calculation of a DAA system could be a better option due to the infinite granularity of Risk Ratios. Alternatively, the use of the Risk Ratio calculation approach for See and Avoid may be a means to allow for "partial credit" for the effects of "See and avoid" when combined with other strategic mitigations as was outlined in Section 3.4;

5. The Probability of Detection for appropriate target types needs to be factored into the determination of Risk Ratio. The approach outlined in this paper underscores the need for manufacturer supplied sensor models that describe the Probability of Detection as a function of range and other target characteristics that are available in the airspace model (e.g., RCS). It is recommended that this requirement be considered as a best practice by groups developing sensor test standards;

6. Airspace models are essential for establishing the DAA system Risk Ratio for the operations in the specific airspace. The airspace models must be region specific to be effective. Furthermore, Section 3.6 highlights the value of airspace models parameterized by aircraft type as target characteristics affect Probability of Detection. The National Research Council of Canada is working in collaboration with TC, Carleton University, and NAV Canada on the development of statistical airspace models for Canadian airspace. These models will include terminal and en-route models, mixed models as well as aircraft types (helicopter, General Aviation, airliner, etc.).

## 5. Conclusions

Many international standards bodies and regulators, including the civilian aviation authority in Canada, Transport Canada (TC), have adopted the SORA approach to guide RPAS operators in their applications for Beyond Visual Line of Sight (BVLOS) flight. The Risk Ratio is a fundamental performance parameter of the tactical mitigations to a mid air collision, and must be evaluated by RPAS operators seeking to operate in ARC-b through ARC-d.

This paper proposed a preliminary, practical, and quantitative approach to determining the Risk Ratio of a given DAA system, considering sensor performance, RPA maneuvering characteristics, airspace specifics, and leveraging publicly available modelling frameworks. An approach for considering the safety gain from "See and avoid" was also presented. Preliminary DAA system parameter sensitivity studies conducted using the proposed approach suggest that the Risk Ratio is more sensitive to sensor FOR than range.

As the approach documented herein was preliminary, and intended for illustrative purposes, it involved several simplifying assumptions. The removal of these assumptions, and accounting for vertical maneuvers, climbing/descending intruders, imperfect detection, and error propagation are the subjects of future work in this area.

**Author Contributions:** Conceptualization, K.E.; Methodology, K.E. and I.B.; Software, K.E. and I.B.; Writing—original draft, K.E. and I.B.; Funding acquisition, I.B. All authors have read and agreed to the published version of the manuscript.

**Funding:** This research received no external funding.

**Data Availability Statement:** The modeling frameworks referenced in this publication are available at: NRC DAA Modelling and Simulation (DAAMSIM) framework (https://github.com/nrc-cnrc/daamsim, accessed on on 3 February 2023), MIT Lincoln Laboratory DAA Evalation of Guidance, Alerting, and Surveillance (DEGAS) framework (https://github.com/mit-ll/degas-surveillance-jwandrews, accessed on 3 February 2023) NASA Detect and AvoID Alerting Logic for Unmanned Systems (DAIDALUS) (https://github.com/nasa/daidalus, accessed on on 3 February 2023).

**Conflicts of Interest:** The authors declare no conflict of interest.

**Abbreviations**

The following abbreviations are used in this manuscript:

| | |
|---|---|
| ADS-B | Automatic Dependent Surveillance—Broadcast |
| ARC | Air Risk Class |
| ATC | Air Traffic Control |
| BVLOS | Beyond Visual Line of Sight |
| CA | Collision Avoidance |
| CPA | Closest Point of Approach |
| CV | Collision Volume |
| DAA | Detect and Avoid |
| DEGAS | DAA Evaluation of Guidance, Alerting, and Surveillance |
| FOR | Field of Regard—The total area perceived by the sensor (FOV plus range) |
| FOV | Field of View—The angular cone perceived by the sensor |
| GA | General Aviation |
| GRC | Ground Risk Class |
| JARUS | Joint Authority on Rulemaking for Unmanned Systems |
| MDPI | Multidisciplinary Digital Publishing Institute |
| MOPS | Minimum Operational Performance Standards |
| NAS | National Airspace System |
| NMAC | Near Mid-Air Collision |
| NRC | National Research Council (of Canada) |
| RCS | Radar Cross Section |
| RPA | Remotely Piloted Aircraft |
| RPAS | Remotely Piloted Aircraft System |
| RR | Risk Ratio |
| RTCA | Radio Technical Commission for Aeronautics |
| RWC | Remain Well Clear |
| SNR | Signal to Noise Ratio |
| SORA | Specific Operations Risk Assessment |
| TC | Transport Canada |
| TMPR | Tactical Mitigation Performance Requirements |
| VLOS | Visual Line of Sight |

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
