# Peer review of "Towards a Quantitative Approach for Determining DAA System Risk Ratio"

_drones, doi:10.3390/drones7020127_

Round 1
Reviewer 1 Report
This paper introduced a quantitative approach for the practitioners to determine the Risk Ratio of a RPA mission, as a complement to the qualitative SORA approach. The approach takes into account the DAA system, RPA performance (in particular the resulted FOR), and airspace specifics.
While the presented work is still at an early stage that relies heavily on simplification assumptions, as indeed claimed by the authors, I believe it provides some added value for the community to develop a more quantitative and consistent approach. In general, the paper is well written and clear to follow.
Some comments and suggestions are as follows:
- It will be helpful to have a high-level description/summary of how the different factors are accounted as a whole, for instance, right after section 3.6. This may also include generalising some other factors that had not been considered in the preliminary work.
- I think the paper will benefit significant if separate numerical experiments can be conducted to showcase the overall Risk Ratio determination process, rather than presenting (potentially independent) illustrative examples at each step.
- The reference format needs some attention.
Author Response
Thank you for your review, comments, and suggestions. We experienced some challenges in addressing the comments owing to a lack of specificity. We suspect that the it wasn't clear enough in our manuscript that the examples provided in Section 3 of the paper consistently used the same RPA characteristics (speed, size, maneuver performance, and avoidance algorithm), based on your comment regarding “presenting (potentially independent) illustrative examples”. To this end we have added text throughout the document that refers the reader back to the system characteristics defined in Section 3.2. Further, we have added clarifying sentences immediately after the example RPAS is introduced (line 291-294 in the revised manuscript) that indicate how the example is used throughout the rest of the section. Owing to space constraints it was not possible to present a wide range of examples with different system characteristics. This will be done in follow-on work, as the focus of this paper was to identify, and detail, the method itself.
Regarding your suggestion that a high level description of how the different factors are accounted for after Section 3.6. We agree that this would add some value, however we have opted to include this high level description of the influence of key factors on the Risk Ratio as a paragraph in the discussion section, as it felt like the most appropriate location. We also added a statement in section 3.6 that indicates that our current airspace model doesn’t support the development of an example that includes probability of detection. This re-enforces our assertion of the importance of appropriate airspace models in the discussion.
You also indicated that the references format requires improvement. We noted one book reference that was not formatted correctly, and have fixed it. Several of our references were technical reports developed by standards/regulatory committees. We are uncertain what the appropriate formatting for the .bib entry for these reports is, and have flagged the entries as @techreport. We have asked the journal for clarification on the preferred .BIB format for tech reports, and we will fix it ASAP upon receipt of a response.
Reviewer 2 Report
The manuscript addresses a very timely and relevant topic. The field of RPAS continues to evolve and while this evolution makes it a challenging to pin down a mitigation strategy for conflicts in BVLOS, this work does a commendable job in addressing the issue. The approach to determining the Risk Ratio only uses a single mode of detection. While this was done in an effort to simplify assumptions, there may be some validity in expanding to include some vertical factors. This was mentioned in the discussion but could be addressed earlier as part of the Risk Ratio discussion.
Author Response
Thank you for your review of our manuscript.
Regarding your comment on multi-mode detection:
As our current ‘mixed’ airspace models do not break down traffic by type, nor by equipage (ADS-B, Mode S, No transponder, etc), we were unable to present an example of multi-mode sensing. We did, however, add some statements in the initial coverage of R0 and Rdet to indicate that this needs to be done on a ‘per sensor’ basis. We hope that this, along with the content in the discussion around line 498 (revised manuscript) addresses the multi-sensor aspect sufficiently. It is our intention to conduct follow up work/publications in this area, likely presenting a more comprehensive analysis of some representative use-cases (i.e. systems and Conops).
Round 2
Reviewer 1 Report
Thank you for revising the paper following my suggestions.